# Effect of Fortification with High-Milk-Protein Preparations on Yogurt Quality

**DOI:** 10.3390/foods14010080

**Published:** 2025-01-01

**Authors:** Justyna Żulewska, Maria Baranowska, Marika Magdalena Bielecka, Aneta Zofia Dąbrowska, Justyna Tarapata, Katarzyna Kiełczewska, Adriana Łobacz

**Affiliations:** Department of Dairy Science and Quality Management, Faculty of Food Science, University of Warmia and Mazury, Oczapowskiego 7, 10-719 Olsztyn, Poland; justyna.zulewska@uwm.edu.pl (J.Ż.); mbb@uwm.edu.pl (M.B.); anetazj@uwm.edu.pl (A.Z.D.); justyna.tarapata@uwm.edu.pl (J.T.); kaka@uwm.edu.pl (K.K.); adriana.lobacz@uwm.edu.pl (A.Ł.)

**Keywords:** casein concentrate, whey concentrate, firmness, sensory evaluation, bacteria count

## Abstract

Protein-enriched yogurts have become increasingly popular among consumers seeking to boost their daily protein intake. The incorporation of milk proteins and protein preparations in yogurt production not only enhances nutritional value but also improves texture, viscosity, and overall sensory properties—key factors that influence consumer acceptance. The main objective of this study was to evaluate the influence of casein and whey protein preparations on the physicochemical properties, viability of lactic acid bacteria, and sensory attributes of yogurts. Yogurts were enriched with 2% (*w*/*w*) protein preparations, including micellar casein preparation (CN85), whey protein isolate (WPI), whey protein concentrate (WPC60), and protein preparations obtained from skim milk by membrane filtration: micellar casein concentrate (CN75) and serum protein concentrate (SPC). The yogurts were produced using the thermostatic method, and their chemical composition, rheological properties, syneresis, firmness, lactic acid bacteria population, and sensory attributes were evaluated. The effects of high-protein preparations derived from skim milk through laboratory-scale membrane filtration processes (SPC, CN75) were compared with those of commercially available protein preparations (SMP, CN85, WPI, and WPC). Obtained results demonstrated that the membrane filtration-derived preparations (SPC and CN75) exhibited advantageous physicochemical properties and supported robust viability of yogurt and probiotic bacteria. However, their sensory quality was marginally inferior compared to the commercial preparations (SMP, CN85, WPI, and WPC). These findings indicate the potential applicability of membrane filtration-derived protein preparations in yogurt production while underscoring the necessity for further investigation to enhance and optimize their sensory characteristics.

## 1. Introduction

Protein-enriched yogurts are gaining popularity among consumers aiming to increase their daily protein intake [1]. In the European Union, milk proteins and protein preparations are not classified as food additives, allowing their incorporation into clean-label food products [2].

Conventionally, syneresis is reduced by increasing the total solids in the yogurt mixture to approximately 14% (*w*/*w*) [3]. In this regard, skimmed milk powder is traditionally used for yogurt production. Casein and whey protein preparations are widely applied in the dairy industry to modify the physicochemical and sensory properties of dairy products, especially yogurts. Enrichment of yogurts with milk proteins enhances their nutritional value, improves texture, and optimizes functional properties, which are critical quality parameters for consumer acceptance. Milk proteins play a significant role in improving the technological functionality of yogurts, including texture (firmness, viscosity, and creaminess) without contributing to increased fat content. In addition, the incorporation of milk protein can reduce the need for stabilizers due to their good water-binding capacity, emulsification, high solubility, gelling, viscosity enhancement, and adhesive interactions that allow obtaining yogurts with a homogeneous structure.

Commercial milk protein preparations are available in various forms, including liquid or powdered, and are characterized by a range of protein concentrations. These preparations include whey protein concentrates (WPCs), whey protein isolate (WPI), micellar casein, and caseinates that improve the texture and functional properties of the final product. Whey protein isolates (WPIs) and concentrates (WPCs) with a protein content of 80% (WPC 80) enable the production of low-lactose yogurts [4]. Whey proteins denatured through heat treatment, used in the preparation of yogurt mixtures, demonstrate exceptionally high water-binding and adhesion capacities. Micellar casein, on the other hand, does not undergo typical thermal denaturation due to the absence of a tertiary structure. However, under conditions of high temperature and low pH, aggregation and coagulation can occur as a result of the mineral balance disruption and partial degradation of casein micelles [5]. The ability to form a gel is one of the key properties of proteins that is crucial for production technology and for shaping the rheological and structural properties of yogurt. The structure of yogurt results from casein aggregation caused by a decrease in pH and the formation of disulfide bonds between κ-casein and denatured whey proteins. Protein content and heat treatment are the most important parameters that determine the textural properties of yogurt. There is evidence of the potential impact of whey proteins on the growth and viability of yogurt bacteria. For example, Glušac et al. [6] observed that the addition of WPC and honey increased the growth and viability of lactic acid bacteria (*Streptococcus thermophilus* and *Lactobacillus delbrueckii* subsp. *bulgaricus*) in yogurt during a 21-day storage period. Similarly, Dąbrowska et al. [7] found that whey protein hydrolysate (WPH) enhanced the growth of yogurt cultures during the early stages of fermentation and contributed to the maintenance of bacterial viability during refrigerated storage. However, WPH did not significantly enhance the growth of *Bifidobacterium adolescentis*, although it notably improved its viability over time. In contrast, Vargas Lopez at al. [8] demonstrated that although WPI can enhance the acid and bile tolerance of *Streptococcus thermophilus* and *Lactobacillus bulgaricus*, it does not consistently promote bacterial growth in yogurt under all conditions. These studies suggest that although whey proteins can influence bacterial viability and fermentation dynamics, their effects may vary depending on the specific formulation and conditions used in yogurt production. Therefore, it is crucial to analyze the impact of protein preparations on the growth of starter cultures as well as on the probiotic *Lactobacillus acidophilus* La-5.

Ongoing advancements in technologies used to fractionate, isolate, and purify milk components provide yogurt producers with enhanced tools for the modification of the composition and properties of yogurt, allowing for better customization to consumer preferences. The growing recognition of milk proteins, especially whey proteins, as valuable food ingredients by consumers worldwide is driving producers to innovate and develop new yogurt varieties. These initiatives require extensive research to confirm the production capabilities and explore innovative product features [1].

The main objective of this study was to evaluate the influence of casein and whey protein preparations, added to milk prior to fermentation, on the physicochemical properties (pH, syneresis, firmness, and viscosity), the viability of lactic acid bacteria (LAB), and the sensory attributes of yogurts. The effects of high-protein preparations derived from skim milk through membrane filtration on a laboratory scale were compared with those of commercially available protein preparations.

The novelty of this research lies in comparing the impact of innovative high-protein preparations derived from skim milk using advanced membrane filtration with commercially available protein preparations. By focusing on these specialized, laboratory-developed ingredients, the study explores their potential to enhance yogurt quality and replace traditional stabilizers, in alignment with clean-label trends and addressing consumer demand for high-protein, natural products.

## 2. Materials and Methods

### 2.1. Materials

Yogurts enriched with 2% (*w*/*w*) protein powder, added prior to fermentation, were prepared using the thermostatic method in a dairy plant at the Department of Dairy Science and Quality Management, University of Warmia and Mazury in Olsztyn (UWM). The following commercial protein preparations were used in the study: micellar casein preparation containing 85% protein content (CN85) (Inleit Ingredients, Curtis-A Coruña, Spain), whey protein isolate containing 91% protein content (WPI) (Superior Ltd., Olsztyn, Poland), whey protein concentrate containing 60% protein content (WPC60) (Superior Ltd., Olsztyn, Poland), as well as protein preparations obtained from skim milk by membrane filtration: micellar casein concentrate containing 75% of protein (CN75) and serum protein concentrate containing 67% protein content (SPC). Membrane filtration was performed at the Department of Dairy Science and Quality Management (UWM) in Olsztyn, Poland.

### 2.2. Experimental Design

Fresh cow’s milk (4.0% fat, 3.4% protein, and 4.86% lactose) was obtained from Holstein-Friesian and Jersey cows maintained in a herd at the Didactic and Research Station at Bałdy University of Warmia and Mazury in Olsztyn (UWM). The milk was heated to 45 °C and skimmed by centrifugation. After normalizing the dry matter content to 11% with skimmed milk powder (SMP), the milk was heated to 45 °C and homogenized at 17.5 MPa. The prepared milk was divided into six portions. High-protein preparations were added to five portions as follows: SPC, CN75, CN85, WPI, and WPC60. SMP was added to the sixth portion at a concentration of 2 g/100 g (control sample) (Figure 1). The milk enriched with protein preparations was thermostated at 45 °C for 40 min.

After pasteurization at 90 °C for 2 min, the milk was cooled to fermentation temperature. Yogurt cultures YF-L811 (Chr. Hansen Poland) and *Lactobacillus acidophilus* La-5^®^ (Chr. Hansen Poland) were added to the milk in the amount recommended by the manufacturer. After being packed into individual containers, fermentation was conducted at 38 °C until reaching a pH of approximately 4.6. The final yogurt was stored refrigerated at 5 ± 1 °C for 21 days and evaluated on days 3 and 21.

### 2.3. Methods

#### 2.3.1. Chemical Composition

The chemical composition of the milk and high-protein yogurts was determined after 3 and 21 days of cold storage using the MilkoScan™ FT2 (Foss, Hillerød, Denmark). The pH of the milk and yogurts was measured using a pH meter (Elmetron CP 551, Zabrze, Poland), calibrated with pH 4.0 and pH 7.0 buffers (POCH, Gliwice, Poland).

#### 2.3.2. Protein Profile

The protein profile of yogurt samples was determined by SDS-PAGE electrophoresis according to the method described by Kiełczewska et al. [9]. Before the analysis, yogurt samples were defatted in the Eppendorf FT15 laboratory centrifuge. Skim yogurt samples were diluted in Laemmli 2× concentrate S3401 sample buffer (Sigma Aldrich, St. Louis, MO, USA), followed by boiling for 5 min at 95 °C to denature the proteins and finally cooled to an ambient temperature. The samples were centrifuged (13,000 rpm, 21 °C, 15 min) and loaded into wells on 4–20% Mini-PROTEAN^®^ TGX™ gel in 15-well plates (Bio-Rad Laboratories Inc., Hercules, CA, USA). The Precision Plus Protein Dual Color Standards (Bio-Rad Laboratories Inc., Hercules, CA, USA) for proteins with a molecular weight of 10–250 kDa were used as the reference bands. The gel was placed in the buffer chamber, and a solution of 10× tris/glycine/SDS running buffer was added (Sigma Aldrich, St. Louis, MO, USA). The gel was initially run at 80 V, with the voltage gradually increasing to 120–150 V as the samples migrated through the gel. Gels were run in the BIO-RAD Mini-protean II cell apparatus (Bio-Rad Laboratories Inc., Hercules, CA, USA). After running, the gels were stained and destained with Coomassie Brilliant Blue. Gel images were captured using the CCD LumiBis imaging system (DNR Bio-Imaging Systems, Modi’in-Maccabim-Re’ut, Israel). The protein content of the samples was determined by densitometry using the TotalLab Quant 1.0 program (TotalLab, Gosforth, UK).

#### 2.3.3. Rheological Properties

The rheological properties of the produced yogurts were tested using the Rheolab QC rotameter by Anton Paar (Graz, Austria) in a coaxial cylinder system with the CC39-SN46605 (CC27/QC-LTD) configuration. The RHEOPLUS/32 program was used to determine the viscosity and shear curves based on the measured values of apparent viscosity and shear stresses over a shear rate range of 1–100 s^−1^. The test was performed at 10 °C with three repetitions for each product after gentle mixing (Rheolab QC rheometer manual).

The results were documented as curves showing the relationship between the averaged values of shear stress and shear rate. The rheological properties of yogurt samples, including the consistency index (K), flow behavior index (n), and apparent viscosity (η) at shear rates of 10, 50, and 100 s^−1^ (η_10_, η_50_, and η_100_) were determined mathematically. The experimental flow curves were approximated using the Ostwald de Waele rheological model, expressed by the following formula:η = Kγ^n−1^
(1)
where η—apparent viscosity [Pa∙s], K—consistency index [Pa·s^n^], γ—shear rate [s^−1^], and n—flow behavior index.

#### 2.3.4. Syneresis of Yogurt

The syneresis of analyzed yogurts was determined using the centrifugal method. A sample of 21 g of yogurt, after mixing, was placed in a calibrated 50 mL test tube and centrifuged for 15 min at 10 °C and 3000× *g* (Thermo Scientific Heraeus Megafuge, Osterode, Germany; 16/16R using TX-400 rotor, 90° Rmax 168 mm), followed by measuring of the separated whey volume. The syneresis was expressed as the percentage of separated whey [10].

#### 2.3.5. Firmness of Yogurt

The firmness of yogurts was evaluated using a texture analyzer TA.XT.plus (Stable Micro Systems Ltd., Godalming, UK). Before measurement, the samples (placed in 100 mL cylindrical cups, dimensions Ø57 × H76 mm) were tempered at 6 °C in a thermal chamber for 4 h. The firmness was measured using a penetration test with a stainless steel cylinder probe (P25, 25 mm diameter). The test parameters were as follows: compression depth of 20 mm; trigger force of 10 g; probe speed before and after compression of 1 and 3 mm/s, respectively. The firmness of the yogurts, defined as the maximum force required to achieve a given deformation, was expressed in Newton [N]. All measurements were performed in at least five replicates.

#### 2.3.6. Microbiological Analysis

MRS (de Man, Rogosa, and Sharpe) agar medium at pH 5.2 was used to selectively enumerate *Lactobacillus delbrueckii* ssp. *bulgaricus* under anaerobic incubation at 45 °C for 72 h. *Lactobacillus acidophilus* La-5 was enumerated using MRS-maltose (MRSM) agar under anaerobic incubation at 43 °C for 72 h. M17 agar medium was used to enumerate *Streptococcus thermophilus* under aerobic conditions at 37 °C for 24 h [11].

Enumeration of bacterial cells was performed by preparing 10-fold serial dilutions of yogurt samples in a sterile bacteriological peptone–water diluent using the pour plate technique. Anaerobic jars and gas-generating kits (Anaerobic System BR 38; Oxoid Ltd., Hampshire, UK) were utilized to create anaerobic conditions. After the appropriate incubation period, plates containing 25 to 250 colonies were counted and recorded as colony-forming units (CFUs) per gram of product, with results expressed as viable counts (log_10_ CFU/g ± SD) in yogurt [8].

#### 2.3.7. Sensory Analysis

The yogurts were sensory analyzed using the profiling method according to EN ISO 13299:2016-05E [12]. The sensory panel included eight assessors previously trained to evaluate dairy products and tested for sensory sensitivity [13]. Based on the elaborated definition sheet, the intensity of 18 sensory attributes and the overall acceptability of the tested yogurts were assessed. The five-point descriptive scale was used in the analysis, where 1 meant the absence of the attribute, and 5 meant the very high intensity of the attribute being evaluated. A total of 6 coded samples were evaluated in the sensory laboratory. Sensory analysis was conducted on the third day after production.

#### 2.3.8. Statistical Analysis

The experiment was carried out in duplicate. The results were analyzed statistically by calculating the mean value and the standard deviation. The results were verified for normal distribution and homogeneity of the variance. Differences were determined by factorial analysis of variance (ANOVA) at a *p* ≤ 0.05 significance level. The Tukey test was used for post hoc analysis. All results were processed in Statistica 13.5 PL software (Statsoft 2017, Krakow, Poland) at *p* ≤ 0.05.

## 3. Results and Discussion

### 3.1. Chemical Composition of Yogurt

The average protein, fat, and lactose content in raw milk was 3.44%, 4.01%, and 4.86%, respectively. The milk was partially skimmed, resulting in a final fat content in the yogurt samples ranging from 0.08% to 0.17%. The addition of powdered high-protein preparations led to an increase in both the protein and lactose content of the yogurt (Table 1).

During the 21-day storage period, a decrease in the lactose content was observed in all yogurt samples, while the protein and fat content showed only minor changes. Consistent with the present study, Jańczuk et al. [14] found that a 28-day refrigeration period did not significantly impact fat content (*p* > 0.05). However, significant differences (*p* ≤ 0.05) were observed in protein and dry matter content depending on the storage duration. In contrast, studies on yogurts made from cow and buffalo milk have shown different results, with dry matter and fat increasing during storage [15].

The chemical composition of the experimental yogurts met the applicable standards [16]. The differences in the nutrient content of the produced yogurts were primarily due to the type of milk protein carrier used. For example, SMP contains more lactose than other milk protein preparations, which resulted in the highest lactose content in the control sample (YSMP), regardless of the storage time. The content of lactic acid in the yogurts after the 3 days of storage ranged from 0.91 to 1.04%, increasing to 1.0–1.14% after the 21 days of storage, as a result of lactose fermentation by yogurt bacteria. This fermentation led to a decrease in the pH of the yogurts over the storage period (Table 1).

Lactic acid fermentation dynamics are crucial in yogurt production, as they affect taste, viscosity, and shelf life. The decrease in milk pH during fermentation triggers protein coagulation through lactic acid production, resulting in the formation of a gel-like structure. Factors such as temperature and incubation time influence yogurt acidification, thereby impacting consistency and flavor [17]. While lactose hydrolysis occurs primarily during the production phase, lactic acid production continues, albeit at a slower rate, during the storage of yogurt, particularly by *L. delbrueckii* subsp. *bulgaricus*. Excessive microbial activity during storage can shorten shelf life, increase acidity, cause syneresis, and affect rheological properties [18].

### 3.2. Reducing Sodium Dodecyl Sulphate Polyacrylamide Gel Electrophoresis (SDS-PAGE)

Densitometric analysis of electrophoretograms enables precise determination of the ratio of whey proteins (WP) to casein (CN) in the yogurt samples (Figure 2). Based on the presented results, it can be concluded that the protein profile of different yogurt types varies depending on the protein preparation used. The YCN85 sample exhibited the lowest ratio of WP to CN, indicating that the addition of 85% micellar casein concentrate significantly increases the proportion of CN relative to WP. Conversely, the highest ratio of WP was observed in YWPI, suggesting that the addition of WPI significantly increases the amount of WP in yogurt. The addition of 75% micellar casein did not significantly affect the WP-to-CN ratio compared to yogurt enriched with milk powder. In summary, the use of different protein preparations significantly influences the protein profile of yogurts. The addition of micellar casein concentrates increases the proportion of CN, while the addition of WPI increases the proportion of WP. The control yogurt and the yogurt with 75% micellar casein exhibited similar WP-to-CN ratios.

In our previous study on non-fermented milk enriched with protein preparations, similar trends regarding the ratio of CN to WP were observed [9]. Specifically, the use of micellar casein concentrates resulted in a higher proportion of CN, whereas the addition of WPI increased the proportion of WP [9]. The observed differences in the casein: whey protein (CN:WP) ratio are particularly relevant for yogurt quality, as this ratio has been shown to significantly influence the rheological properties, such as storage modulus (G’), yield stress, and yield strain of low-fat yogurt [19]. The study by Zhao et al. [19] demonstrated that a decrease in the CN:WP ratio leads to a denser and more finely cross-linked network, with smaller, more evenly distributed pores, resulting in a more homogeneous yogurt structure when the CN:WP ratio is approximately 1:1, compared to higher ratios such as 4:1 or 3:1. Chua et al. [20] emphasized that modifying the CN:WP ratio significantly influences the textural, rheological, microstructural, and aroma compound release properties of non-fat stirred yogurts. Their findings showed that yogurts with lower CN:WP ratios tend to be firmer and stiffer.

### 3.3. Rheological Properties

An analysis of the flow curves showed that the changes in shear stress values for most yogurts, within the shear rate range of 1 s^−1^ to 15.10–18.7 s^−1^ followed a similar pattern (Figure 3 and Figure 4). At higher shear rates, all curves exhibited a flattening trend. At shear rates above 20 s^−1^, the changes in shear stress for most yogurt samples were minimal.

The shape and characteristics of the flow curves for the samples evaluated on the 21st day of storage were consistent with those observed on the 3rd day. In contrast, shear stress values decreased for the YSMP, YCN85, YWPC, and YCN75 samples, showed minimal change for the YSPC sample, and increased for yogurts with added WPI (Figure 4).

As the shear rate increases, the viscosity decreases due to the disruption of the gel structure. The higher the shear rate, the more pronounced the reduction in viscosity. The apparent viscosity continues to decrease until further alignment of the particles along the flow direction is no longer feasible, at which point the flow and viscosity curve transitions into a linear relationship [21].

The calculated apparent viscosity of all yogurts was highest and most variable at a shear rate of 10 s^−1^. All calculated rheological parameters and shear stresses for yogurts with WPI addition were characterized by significantly higher values compared to the other samples (Figure 3 and Figure 4, Table 2). YWPC yogurts exhibited the lowest viscosity, with values comparable to those of samples with SMP, ranging from 5.3 to 6.5 Pa∙s. The addition of other high-protein preparations (CN75, CN85, and SPC) increased the viscosity of the yogurts compared to those mentioned above, although no significant differences were observed between these preparations. The viscosity of the YCN75, YCN85, and YSPC yogurts ranged from 7.3 to 8.8 Pa∙s (Table 2).

As the shear rate increased, the viscosity of the yogurts decreased, and the differences between samples became less pronounced. At a shear rate of 50 s^−1^, the viscosity of most samples ranged from 1.25 to 2.10 Pa∙s, with the samples containing WPI exhibiting higher viscosities (3.0–3.3 Pa∙s). At a shear rate of 100 s^−1^, the viscosity was lower, with the smallest variation, ranging from 0.67 to 1.1 Pa∙s. The viscosity of YWPI samples at this shear rate was the highest (1.55–1.67 Pa∙s). During storage, the viscosity of the yogurts containing WPI increased, whereas the viscosity of the other samples decreased, irrespective of the shear rate.

According to Isleten and Karagul-Yuceer [22], the addition of WPI to non-fat yogurts resulted in the greatest increase in viscosity, possibly due to a reduction in CN relative to WP [23].

Replacing SMP with WPC increases the gelation strength and viscosity of yogurt [23]. A study by Królczyk et al. [24] showed that the addition of WPC34 or WPC80 at 0.7–2.0% and 0.5–0.8%, respectively, is sufficient to improve the yogurt curd characteristics. However, increasing the concentration of these additives may negatively impact their sensory characteristics. The flow index (n), which characterizes the non-Newtonian fluid behavior of the yogurt, was less than 1 for all samples (Table 2), indicating shear-thinning behavior typical of yogurts [25]. The consistency coefficient (K), which is a measure of the viscosity of the tested sample, varied depending on the type of protein preparation in the yogurt and storage time (Table 2).

The yogurt with added milk SPC showed the most stable rheological properties during the 21-day storage period. The values of shear stress and the calculated apparent viscosity and other rheological coefficients (n, K) showed the least fluctuations during storage.

The apparent viscosity determined at shear rates of 10 s^−1^, 50 s^−1^, and 100 s^−1^ corresponds to rheological changes in the yogurt curd during consumption. Shear rates between 50 s^−1^ and 100 s^−1^ correspond to chewing and swallowing actions [26]. Ningtyas et al. [27] showed that at a shear rate of 50 s^−1^, the apparent viscosity of acid-rennet cheese correlates with changes in the consistency of the product and the mouthfeel during consumption. Vidigal et al. [28] found that the density of semi-solid products in the mouth responds to the value of apparent viscosity at a shear rate of 10 s^−1^.

### 3.4. Firmness of Yogurts

During storage, the firmness of the yogurt samples increased except for the YWPI sample. The highest firmness was observed in the yogurt with the addition of WPI (4.06 N after the 3rd and 3.92 N after the 21st day of storage) and the lowest in YWPC (1.45 N after the 3rd day of storage) (Figure 5). The yogurt gel with WPI was very firm, which can be attributed not only to the overall protein content—the highest among the tested yogurts (Table 1)—but also to the functional properties of whey proteins, such as high water-holding capacity. The very dense curd suggests that the protein matrix was tightly packed, requiring higher shear stresses to disrupt the gel structure of the YWPI yogurt. No significant difference was observed between the firmness of yogurt containing WPC and the control samples with the addition of skimmed milk powder (SMP) after 3 and 21 days (*p* > 0.05) (Figure 5). However, the addition of other protein components (CN75, CN85, and SPC) increased the firmness of the yogurt compared to the reference samples (*p* ≤ 0.05), although to a lesser extent than with WPI.

Similar results were reported by Skrzypczak and Gustaw (2012), who showed that the firmness of WPI-enriched bio-yogurts was higher than that of those enriched with SMP [29]. Syneresis after 21 days of storage in YSMP and YSPC yogurts was greater than that observed after 3 days (*p* ≤ 0.05). The curd in the remaining stored samples could retain more whey than the yogurts after 3 days. After 21 days, the lowest syneresis was observed in YWPI samples (35.97%). The greatest reduction in syneresis was caused by the enrichment of yogurt in WPI (*p* ≤ 0.05). At the same time, casein concentrates (CN75 and CN85) significantly decreased the degree of syneresis (*p* ≤ 0.05), but to a lesser extent (Figure 6).

The yogurt curd containing WPC was the least susceptible to syneresis. After 3 and 21 days of storage, these samples released 45.41 and 43.69% whey during centrifugation, respectively (Figure 6). The high water-binding capacity of whey proteins contributed to the enhanced stability of these curds. This finding aligns with studies by Akalın et al. [30], which demonstrated that WPC, after thermal treatment, improved the water-holding capacity and reduced syneresis in yogurt by enhancing the bridging between protein particles. Sodini et al. [31] investigated the effect of WPC on the properties of yogurts and found that the water holding capacity was higher in samples enriched with WPC than in yogurt with skimmed milk powder addition.

### 3.5. Microbiological Analysis

The initial counts of particular starter bacteria varied and were not dependent on the type of protein carrier used. The results of the evaluation conducted on the 3rd day after production indicate that the smallest increase was observed in the number of *L. delbruecki* ssp. *bulgaricus*. The largest increase in the count of these bacteria, from 5.99 log to 7.32 log, was found in yogurts with YWPC. A similar increase in the number of *L. delbruecki* ssp. *bulgaricus* was observed in YSPC samples (Table 3).

The greatest increase in bacterial counts during fermentation and 3 days of storage was found for *L. acidophilus*, with counts rising from 5.1–5.9 log to 7.3–8.2 log. The greatest increase in the count of *L. acidophilus* La-5 was observed in yogurts with SPC and WPC.

The highest number of bacteria, regardless of the type of protein preparation and storage time, was observed for *S. thermophilus*. The initial number of these bacteria ranged from 6.94 to 7.31 log, increased to 8.76–9.90 log after 3 days, and then decreased to 8.64–9.54 log after 21 days (Table 3). The highest number of these bacteria was found in the yogurt containing YCN75 after 3 days of storage. After 21 days of storage, the viability of starter bacteria in yogurts varied and depended on both the type of bacteria and storage time, as well as the type of protein preparation. Throughout the storage period, *S. thermophilus* bacteria were the most numerous and stable over time. On the other hand, the greatest decrease was observed in the number of *L. acidophilus*; however, no type of yogurt exhibited a bacterial count lower than 10^6^ CFU/g, which is the minimum threshold for additional bacteria specified in the Codex Standard for Fermented Milks [16].

In the present study, the total number of yogurt starter bacteria after 21 days of storage met the minimum requirement of the Codex Standard for Fermented Milks, set at 10^7^ CFU/g. In the tested yogurts, the total count of yogurt bacteria ranged from 8.67 to 9.55 log [16].

When evaluating the effect of individual protein preparations on the number of starter and additional bacteria, it can be concluded that the addition of whey protein more effectively promoted the growth and survival of these bacteria compared to casein protein. Among the whey protein concentrates used, WPI had the least favorable effect.

The growth and survival of lactic acid bacteria (LAB) in fermented milks are influenced by various factors, including bacterial species and type, fermentation time and temperature, storage conditions, acidity, and dry matter content, including carbon and nitrogen sources. Bacterial viability is also affected by nutrient availability during storage and the presence of oxygen [32]. The results obtained show a loss or reduction in the viability of both yogurt and probiotic bacteria over the storage period, which may be attributed to post-acidification during storage [33].

### 3.6. Sensory Analysis

The sensory analysis evaluated five attributes of the yogurts: appearance, aroma, consistency, mouthfeel, and taste (Table 4). Overall acceptability was also assessed.

The sensory analysis results of appearance indicated that all the tested yogurts were white, slightly creamy, and very uniform in color. There was no negative effect of the high-protein preparations on the color, as often reported by other authors [34,35]. Some differentiation in whey separation was observed, being most evident in YSMP (3.8) and YCN75 (3.5) samples and least noticeable in YWPC and YCN85 yogurts. The hypothesis that the addition of high-protein preparations reduces whey separation in fermented milks was partially confirmed [36].

The tested yogurts differed in overall aroma intensity and in a typical yogurt aroma (*p* ≤ 0.05). The control sample had the highest overall aroma intensity and the strongest yogurt aroma, while in the other yogurts, these characteristics were detectable at similar intensity levels.

In the evaluation of consistency attributes, no statistically significant differences were found regarding uniformity, thickness, and viscosity. Yogurt with SPC was distinguished by having higher ratings of lumpiness.

Assessing attributes related to the mouthfeel, it was found that yogurts with WPC and WPI were distinguished by slightly less adhesion to the palate than the other samples, but the differences were not significant. Statistically significant differences were noted for smoothness and mealiness. The smoothest mouthfeel characterized YWPC and YCN85 yogurts, while the moderate mealy mouthfeel was detectable in YSPC and YCN75.

In the taste evaluation, statistically significant differences were found in overall taste intensity, typical yogurt taste, sour taste, and atypical taste. The control sample had the highest overall taste intensity and the most typical yogurt taste. YSPC yogurt was the most sour, while YWPI was the least sour. Four samples exhibited a minimally detectable atypical taste, with the most noticeable in YCN75 yogurt, which was also slightly bitter. The bitter taste sensation may indicate the release of casein-derived peptides [37].

In terms of overall acceptability, YCN85 yogurt received the highest scores, while YCN75 yogurt received the lowest scores. Yogurts with protein preparations obtained from skimmed milk by membrane filtration—micellar casein concentrate (YCN75) and serum protein concentrate (YSPC)—were scored significantly lower than control yogurt in overall acceptability.

Yogurt with micellar casein preparation with 85% protein content (YCN85) and yogurt with whey protein concentrate (YWPC) received the best sensory ratings. The product labeled as YCN75 was scored lower due to a slightly bitter taste and mealiness. In turn, YWPI had a less uniform consistency than the others. Yogurt with milk serum protein concentrate (YSPC) had the most noticeable lumpiness.

## 4. Conclusions

This study demonstrates that the type of high-protein preparation significantly influences the physicochemical, microbial, and sensory properties of yogurt. Whey protein isolate (WPI) resulted in the highest yogurt firmness and the lowest degree of syneresis. Whey protein preparations, particularly serum protein concentrate (SPC), exhibited superior water-binding capacity compared to casein preparations, leading to reduced syneresis and improved yogurt texture.

All samples exhibited non-Newtonian, pseudoplastic flow characteristics, indicating that, despite differences in protein composition and content, the yogurts maintained consistent flow properties. The most stable rheological properties were observed in yogurt containing SPC. Although refrigerated storage affected yogurt acidity, firmness, and syneresis, the choice of protein preparation had a more pronounced impact than storage time alone.

Despite favorable physicochemical attributes and good viability of yogurt bacteria, including *Lactobacillus acidophilus* La-5, the preparations obtained at the laboratory scale (CN75 and SPC) exhibited slightly lower sensory quality compared to commercial formulations (WPI, WPC, and CN85), suggesting the need for further optimization.

This study provides valuable insights into the physicochemical properties, lactic acid bacteria (LAB) viability, and sensory characteristics of yogurts enriched with membrane-filtered proteins. Overall, this research contributes to the development of healthier, high-protein dairy products. Membrane filtration offers a sustainable and cost-effective method for protein concentration, holding significant potential for the dairy industry, particularly in response to increasing consumer interest in protein-enriched dairy products.

## Figures and Tables

**Figure 1 foods-14-00080-f001:**
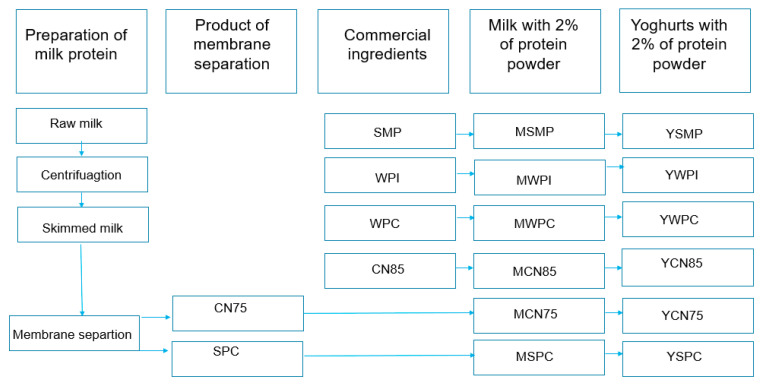
The production scheme for yogurts enriched with high-protein preparations. SMP—skim milk powder with 35% protein, WPI—whey protein isolate with 91% protein content, WPC—whey protein concentrate with 60% protein content, CN85—micellar casein preparation with 85% protein content, CN75—micellar casein concentrate with 75% protein content, SPC—serum protein concentrate with 67% protein content, M/YSMP—milk/yogurt with skim milk powder with 35% protein content, control sample, M/YWPI—milk/yogurt with whey protein isolate, M/YWPC—milk/yogurt with whey protein concentrate with 60% protein content, M/YCN85—milk/yogurt with micellar casein concentrate with 85% protein content, M/YCN75—milk/yogurt with micellar casein concentrate with 75% protein content, M/YSPC—milk/yogurt with serum protein with 67% protein content.

**Figure 2 foods-14-00080-f002:**
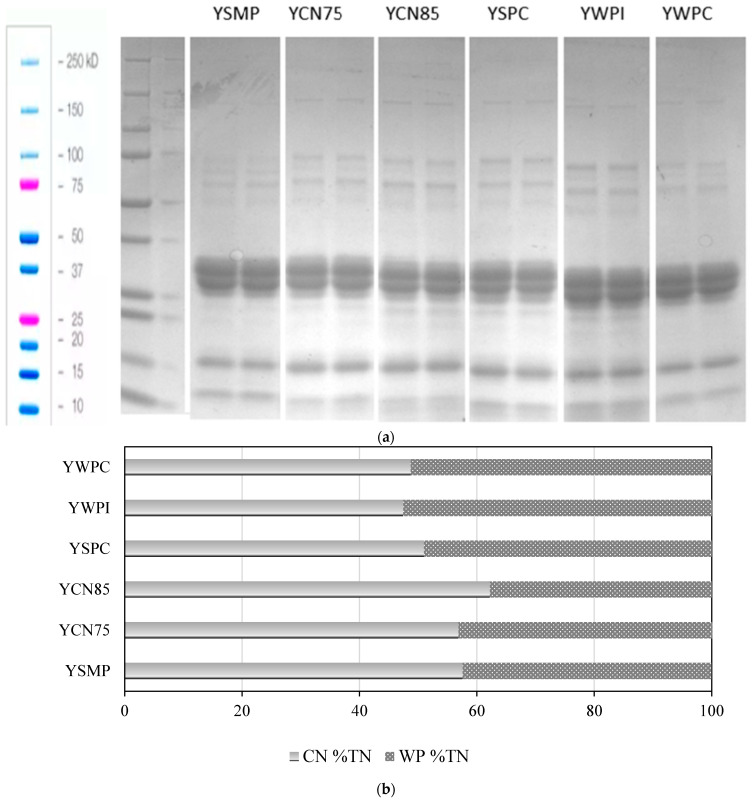
(**a**) SDS-PAGE electrophoretogram of yogurt samples. (**b**). Average content of casein and whey proteins in a densitometric analysis of electrophoretograms (n = 2). CN—casein, WP—whey protein, TN—total nitrogen, YSMP—yogurt with skim milk powder with 35% protein content, control sample, YCN75—yogurt with micellar casein concentrate with 75% protein content, YCN85—yogurt with micellar casein concentrate with 85% protein content, YSPC—yogurt with serum protein with 67% protein content, YWPI—yogurt with whey protein isolate, and YWPC—yogurt with whey protein concentrate with 60% protein content.

**Figure 3 foods-14-00080-f003:**
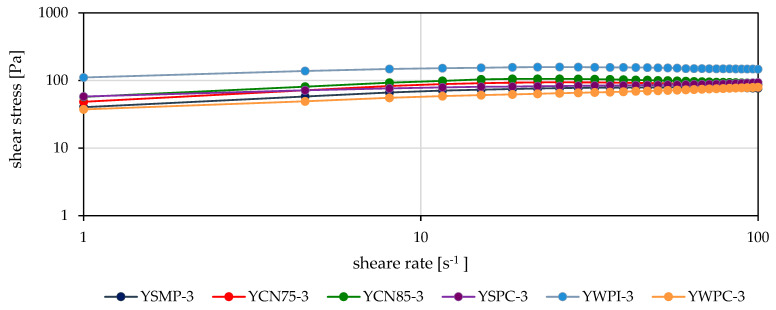
Flow curves of experimental yogurts after 3 days of storage.

**Figure 4 foods-14-00080-f004:**
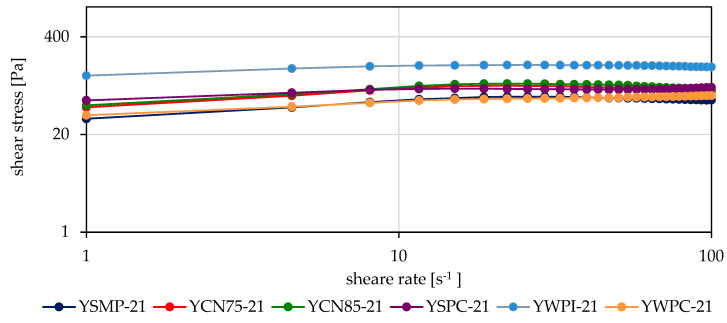
Flow curves of experimental yogurts after 21 days of storage.

**Figure 5 foods-14-00080-f005:**
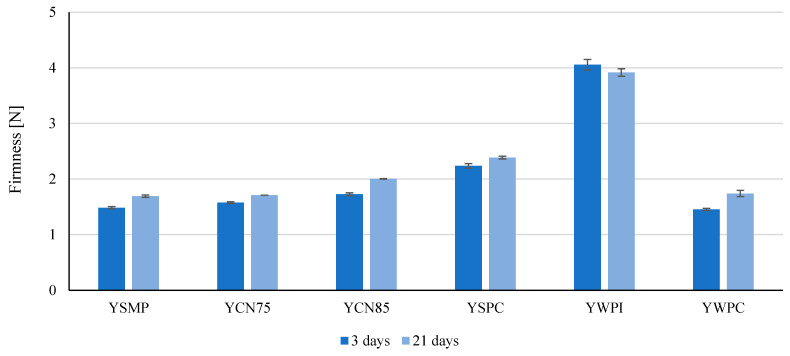
Firmness of yogurts enriched with 2% *w*/*w* of high-protein preparations after 3rd and 21st days of storage. Values are means ± SD (n = 6). YSMP—yogurt with skim milk powder with 35% protein content, control sample, YCN75—milk/yogurt with micellar casein concentrate with 75% protein content, YCN85—yogurt with micellar casein concentrate with 85% protein content, YSPC—yogurt with serum protein with 67% protein content, YWPI—yogurt with whey protein isolate, and YWPC—yogurt with whey protein concentrate with 60% protein content.

**Figure 6 foods-14-00080-f006:**
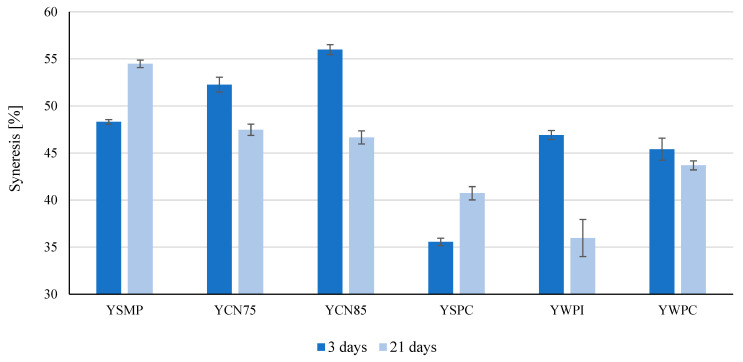
Syneresis of yogurts enriched with 2% *w*/*w* of high-protein preparations after 3rd and 21st days of storage. Values are means ± SD (n = 6). YSMP—yogurt with skim milk powder with 35% protein content, control sample, YCN75—yogurt with micellar casein concentrate with 75% protein content, YCN85—yogurt with micellar casein concentrate with 85% protein content, YSPC—yogurt with serum protein with 67% protein content, YWPI—yogurt with whey protein isolate, and YWPC—yogurt with whey protein concentrate with 60% protein content.

**Table 1 foods-14-00080-t001:** Chemical composition of yogurts with high-protein preparations after the 3rd and 21st day of storage.

Yogurts	Storage Time	Fat	Protein	Lactose	Dry Matter	Lactic Acid	pH
	Days	g/100 g					
YSMP	3	0.17 ^d^ ± 0.02	4.68 ^c^ ± 0.02	6.54 ^g^ ± 0.04	12.79 ^d^ ± 0.06	1.00 ^b^ ± 0.01	4.57 ^a^ ± 0.01
	21	0.14 ^cd^ ± 0.01	4.73 ^c^ ± 0.01	6.39 ^f^ ± 0.08	12.78 ^d^ ± 0.11	1.11 ^d^ ± 0.01	4.39 ^c^ ± 0.01
YMCN75	3	0.10 ^ac^ ± 0.02	5.34 ^a^ ± 0.02	5.49 ^abc^ ± 0.07	12.75 ^abc^ ± 0.01	1.04 ^a^ ± 0.01	4.58 ^a^ ± 0.005
	21	0.08 ^a^ ± 0.02	5.3 ^ab^ ± 0.04	5.42 ^ab^ ± 0.04	12.52 ^aef^ ± 0.09	1.14 ^e^ ± 0.02	4.44 ^b^ ± 0.001
YCN85	3	0.09 ^a^ ± 0.01	5.28 ^f^ ± 0.01	5.58 ^cd^ ± 0.01	12.91 ^abc^ ± 0.11	1.01 ^bc^ ± 0.00	4.63 ^d^ ± 0.01
	21	0.07 ^a^ ± 0.02	5.37 ^a^ ± 0.02	5.53 ^abc^ ± 0.03	12.87 ^abc^ ± 0.18	1.13 ^de^ ± 0.00	4.43 ^b^ ± 0.002
YSPC	3	0.10 ^ac^ ± 0.02	5.33 ^ab^ ± 0.02	5.49 ^abc^ ± 0.07	12.75 ^abf^ ± 0.15	1.04 ^a^ ± 0.01	4.57 ^a^ ± 0.000
	21	0.08 ^a^ ± 0.02	5.30 ^b^ ± 0.04	5.42 ^a^ ± 0.04	12.52 ^ef^ ± 0.09	1.08 ^f^ ± 0.01	4.41 ^c^ ± 0.017
YWPI	3	nd ^b^	5.52 ^e^ ± 0.07	5.54 ^bc^ ± 0.03	12.78 ^d^ ± 0.09	0.91 ^g^ ± 0.01	4.56 ^a^ ± 0.008
	21	nd ^b^	5.65 ^e^ ± 0.09	5.41 ^ab^ ± 0.00	12.68 ^bcd^ ± 0.07	1.02 ^ac^ ± 0.00	4.44 ^a^ ± 0.005
YWPC	3	nd ^b^	5.15 ^d^ ± 0.27	5.96 ^e^ ± 0.02	12.58 ^abc^ ± 0.04	0.95 ^h^ ± 0.00	4.56 ^a^ ± 0.005
	21	nd ^b^	5.10 ^d^ ± 0.00	5.78 ^d^ ± 0.04	12.37 ^f^ ± 0.07	1.05 ^a^ ± 0.01	4.44 ^b^ ± 0.005

Values are means ± SD (n = 6). Values with different superscripts in columns differ significantly at *p* ≤ 0.05. YSMP—yogurt with skim milk powder with 35% protein content, control sample, YCN75—yogurt with micellar casein concentrate with 75% protein content, YCN85—yogurt with micellar casein concentrate with 85% protein content, YSPC—yogurt with serum protein with 67% protein content, YWPI—yogurt with whey protein isolate, YWPC—yogurt with whey protein concentrate with 60% protein content, and nd—not detected.

**Table 2 foods-14-00080-t002:** Characteristics of rheological properties of yogurt samples after 3 and 21 days of storage.

Yogurt Samples	Time [Days]	Parameters of Ostwald-de Waele Rheological Model			Apparent Viscosity [Pa·s]		
		K [Pa∙s^n^]	N	R^2^	η_10_	η_50_	η_100_
YSMP	3	49.51 ± 0.42	0.115 ± 0.005	0.818	6.45	1.55	0.84
	21	41.34 ± 0.79	0.106 ± 0.003	0.608	5.28	1.25	0.67
YCN75	3	62.34 ± 0.20	0.098 ± 0.006	0.581	7.81	1.83	0.98
	21	59.97 ± 0.70	0.088 ± 0.009	0.488	7.34	1.69	0.90
YCN85	3	70.72 ± 0.51	0.097 ± 0.008	0.587	8.83	2.06	1.10
	21	60.33 ± 0.16	0.110 ± 0.001	0.603	7.77	1.86	1.00
YSPC	3	63.92 ± 0.63	0.088 ± 0.003	0.954	7.49	1.73	0.92
	21	65.49 ± 0.57	0.054 ± 0.004	0.657	7.42	1.62	0.84
YWPI	3	131.52 ± 0.56	0.0356 ± 0.07	0.339	14.28	3.02	1.55
	21	143.21 ± 0.72	0.0347 ± 0.09	0.355	15.51	3.28	1.68
YWPC	3	38.63± 0.66	0.154 ± 0.009	0.992	5.50	1.41	0.78
	21	40.46 ± 0.48	0.111 ± 0.032	0.911	5.22	1.25	0.67

Values are means ± SD (n = 6). K—consistency coefficient; n—flow behavior index; R^2^—coefficient of determination; η_10_, η_50_, and η_100_—apparent viscosity at shear rate 10, 50, and 100 s^−1^, respectively; YSMP—milk/yogurt with skim milk powder, control sample with 35% protein content; YCN75—milk/yogurt with micellar casein concentrate with 75% protein content; YCN85—milk/yogurt with micellar casein concentrate with 85% protein content; YSPC67—milk/yogurt with serum protein with 67% protein content; YWPI—milk/yogurt with whey protein isolate; YWPC60—milk/yogurt with whey protein concentrate with 60% protein content.

**Table 3 foods-14-00080-t003:** The count of yogurt bacteria *Lactobacillus delbrueckii* ssp. *bulgaricus* and *Streptococcus thermophilus* and probiotic bacteria *Lactobacillus acidophilus* La-5.

Sampling Time	YSMP	YCN75	YCN85	YSPC	YWPI	YWPC
	Log CFU/g					
*Lactobacillus delbrueckii* ssp. *bulgaricus*						
before fermentation	5.96 ± 0.02	6.18 ± 0.12	6.70 ± 0.01	6.59 ± 0.01	6.99 ± 0.05	5.99 ± 0.05
3 days	6.19 ± 0.17	6.58 ± 0.07	6.79 ± 0.01	7.16 ± 0.02	6.35 ± 0.40	7.29 ± 0.18
21 days	7.49 ± 0.21	7.49 ± 0.09	5.22 ± 0.01	8.11 ± 0.05	7.53 ± 0.06	7.73 ± 0.07
*Streptococcus thermophilus*						
before fermentation	7.24 ± 0.03	6.94 ± 0.02	7.17 ± 0.16	7.23 ± 0.05	7.30 ± 0.05	7.25 ± 0.02
3 days	9.32 ± 0.02	9.89 ± 0.04	9.09 ± 0.05	8.75 ± 0.09	9.45 ± 0.01	9.40 ± 0.01
21 days	8.64 ± 0.09	8.90 ± 0.14	9.20 ± 0.05	9.07 ± 0.03	9.44 ± 0.04	9.54 ± 0.09
Total count of bacteria *Lactobacillus delbrueckii* ssp. *bulgaricus* and *Streptococcus thermophilus*						
21 days	8.67	8.93	9.20	9.10	9.44	9.55
*Lactobacillus acidophilus* La-5						
before fermentation	5.32 ± 0.14	5.14 ± 0.15	5.08 ± 0.18	5.47 ± 0.09	5.52 ± 0.19	5.58 ± 0.24
3 days	7.24 ± 0.26	7.65 ± 0.27	7.74 ± 0.18	8.14 ± 0.11	6.65 ± 0.33	8.22 ± 0.14
21 days	8.66 ± 0.03	8.63 ± 0.11	8.89 ± 0.13	8.41 ± 0.01	6.63 ± 0.06	7.62 ± 0.02

Values are means ± SD (n = 6). YSMP—yogurt with skim milk powder with 35% protein content, control sample, YCN75—yogurt with micellar casein concentrate with 75% protein content, YCN85—yogurt with micellar casein concentrate with 85% protein content, YSPC67—yogurt with serum protein with 67% protein content, YWPI—yogurt with whey protein isolate, and YWPC60—yogurt with whey protein concentrate with 60% protein content.

**Table 4 foods-14-00080-t004:** Mean values of sensory attributes of yogurts with high protein preparations (n = 8).

Sensory Attributes	YSMP	YCN75	YCN85	YSPC	YWPI	YWPC	*p*-Value
Appearance							
creamy color	3.0	3.0	2.8	2.8	2.8	2.8	>0.05
uniform color	4.5	4.4	4.4	4.4	4.4	4.4	>0.05
whey separation	3.8 ^c^	3.5 ^c^	2.5 ^a^	3.0 ^b^	3.0 ^b^	2.5 ^a^	0.000
Aroma							
overall intensity	4.3 ^c^	3.8 ^b^	3.1 ^a^	3.5 ^ab^	3.5 ^ab^	3.5 ^ab^	0.002
typical for yogurt	4.4 ^b^	3.8 ^a^	3.5 ^a^	3.8 ^a^	3.8 ^a^	3.8 ^a^	0.044
atypical	1.0	1.0	1.0	1.0	1.0	1.0	>0.05
Consistency							
uniform	4.0	4.3	4.1	4.0	3.9	4.0	>0.05
lumpy	2.1 ^b^	2.3 ^b^	1.3 ^a^	3.4 ^c^	2.0 ^b^	1.9 ^ab^	0.000
thick	3.4	3.8	3.8	3.8	3.6	3.6	>0.05
viscous	2.6	2.5	2.6	3.0	2.3	2.5	>0.05
Mouthfeel							
adhesive	2.1	2.1	2.3	2.3	1.9	1.8	>0.05
smooth	3.1 ^b^	3.1 ^b^	3.9 ^c^	2.4 ^a^	2.1 ^a^	4.1 ^c^	0.002
mealy	1.9 ^b^	2.3 ^c^	1.9 ^b^	2.3 ^c^	1.1 a	2.0 ^bc^	0.000
Taste							
overall intensity	4.0 ^c^	3.4 ^b^	3.3 ^b^	3.3 ^b^	2.6 ^a^	3.4 ^b^	0.000
typical for yogurt	4.0 ^c^	3.4 ^bc^	3.4 ^bc^	3.1 ^ab^	2.6 ^a^	3.6 ^c^	0.003
sour	3.4 ^ab^	3.0 ^ab^	3.1 ^ab^	3.6 ^b^	2.8 ^a^	3.1 ^ab^	0.011
bitter	1.0	1.3	1.0	1.0	1.0	1.0	>0.05
atypical	1.5 ^ab^	1.9 ^b^	1.1 ^a^	1.5 ^ab^	1.5 ^ab^	1.1 ^a^	0.018
overall acceptability	3.5 ^bc^	2.4 ^a^	4.3 ^c^	2.8 ^a^	3.1 ^ab^	3.8 ^bc^	0.002

^a,b,c^—Mean values in a row with different letters are significantly different at *p* ≤ 0.05, YSMP—yogurt with skim milk powder with 35% protein conent, control sample, YCN75—yogurt with micellar casein concentrate with 75% protein content, YCN85—yogurt with micellar casein concentrate with 85% protein content, YSPC67—yogurt with serum protein with 67% protein content, YWPI—yogurt with whey protein isolate, and YWPC60—yogurt with whey protein concentrate with 60% protein content.

## Data Availability

The original contributions presented in the study are included in the article, further inquiries can be directed to the corresponding author.

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
