# Peer review of "Effect of Fortification with High-Milk-Protein Preparations on Yogurt Quality"

_foods, 2025, doi:10.3390/foods14010080_

Round 1
Reviewer 1 Report
Comments and Suggestions for Authors
The paper is now fine
Author Response
Dear Reviewer,
Thank you very much for your review and valuable feedback. We appreciate your thoughtful comments and are pleased that you found the paper satisfactory. Your insights have been helpful in improving the quality of our work.
Kind regards,
Marika Bielecka

Reviewer 2 Report
Comments and Suggestions for Authors
The novelty of this research lies in comparing the impact of innovative high-protein preparations derived from skim milk via advanced membrane filtration with commercially available protein preparations. This study explores the potential of high milk-derived protein preparations to improve yoghurt quality and replace traditional stabilisers in response to clean label trends and consumer demand for high-protein natural products. However, in my opinion, there are some comments that need to be addressed to improve the quality of the manuscript and make it acceptable for publication. Therefore, according to following points, some modifications are needed:
The authors need to clarify in the introduction section the significant contribution of this study to the food industry and even the dairy industry.
The authors are advised to describe in detail the method of 2.3.2. Reducing-sodium dodecyl sulphate polyacrylamide gel electrophoresis (SDS-PAGE).
Whether the p in “p<0.05” and “P ≤ 0.05” are upper or lower case should be consistent and should also be italicized.
Lines 200-202: It is recommended that the authors delete “This section may be divided by subheadings. It should provide a concise and precise description of the experimental results, their interpretation, as well as the experimental conclusions that can be drawn.”
Line 221: The author needs to add a reference to this sentence.
The authors must add a discussion of the relevant results in “3.2. Reducing-sodium dodecyl sulphate polyacrylamide gel electrophoresis (SDS-PAGE)”.
Line 306: [14 Puvanenthiran et al. 2002], Line 324: Food Hydrocolloids, 84, 173-180.], Line 374: [21. Sodini et al. (2005) Please standardize the format of references.\
Line 420: [24???
Please simplify the conclusions section by showing the main conclusions, not a repetition of results, much less a discussion.
There are too few references, and it is recommended that more references be added to improve the persuasiveness of the paper's discussion.
Please add DOI in references 3, 12, 20, 26.
Comments on the Quality of English LanguageThe English could be improved to more clearly express the research.
Author Response
Dear Reviewer,
Thank you very much for your detailed review and valuable comments on our manuscript. We sincerely appreciate your constructive feedback, which has helped us improve the quality and clarity of our work. Below, we address your comments point by point:
- Clarification in the Introduction Section:
Comment: "The authors need to clarify in the introduction section the significant contribution of this study to the food industry and even the dairy industry."
Response: We have clarified the contribution of this study to the food and dairy industries by emphasizing how membrane filtration can improve yogurt quality and meet consumer demand for high-protein, clean-label products. Additionally, we highlight the sustainability and cost-effectiveness of this method.
- Description of SDS-PAGE Method (Section 2.3.2):
Comment: "The authors are advised to describe in detail the method of 2.3.2. Reducing-sodium dodecyl sulphate polyacrylamide gel electrophoresis (SDS-PAGE)."
Response: We have expanded Section 2.3.2 to provide a detailed description of the SDS-PAGE method, ensuring clarity and reproducibility. The revised section now includes specifics on sample preparation, electrophoresis conditions, and imaging procedures.
- Consistency in “p < 0.05” Notation:
Comment: "Whether the p in 'p < 0.05' and 'P ≤ 0.05' are upper or lower case should be consistent and should also be italicized."
Response: We have ensured that the notation is consistent throughout the manuscript, using lowercase p and italicizing it appropriately.
- Removal of Placeholder Text (Lines 200-202):
Comment: "It is recommended that the authors delete ‘This section may be divided by subheadings...’"
Response: This sentence has been removed as recommended.
- Addition of References (Line 221):
Comment: "The author needs to add a reference to this sentence."
Response: A relevant reference has been added to support the statement in Line 221.
- Discussion in Section 3.2 (SDS-PAGE Results):
Comment: "The authors must add a discussion of the relevant results in ‘3.2. Reducing-sodium dodecyl sulphate polyacrylamide gel electrophoresis (SDS-PAGE).’"
Response: We have added a discussion of the SDS-PAGE results in Section 3.2, including trends in the casein-to-whey protein ratio and their impact on yogurt quality. We referenced related studies to contextualize our findings.
- Standardization of References (Lines 306, 324, 374, 420):
Comment: "Please standardize the format of references."
Response: The references have been standardized to ensure consistency in formatting.
- Simplification of the Conclusions Section:
Comment: "Please simplify the conclusions section by showing the main conclusions, not a repetition of results."
Response: The conclusions section has been revised to focus on the primary findings and implications, avoiding repetition of the results.
- Addition of References:
Comment: "There are too few references, and it is recommended that more references be added to improve the persuasiveness of the paper's discussion."
Response: We have incorporated additional references to strengthen the discussion and provide a more comprehensive context for our findings.
- DOI for References 3, 12, 20, and 26:
Comment: "Please add DOI in references 3, 12, 20, 26."
Response: We have checked these references, but DOI numbers are not available for them.
- Language Quality:
Comment: "The English could be improved to more clearly express the research."
Response: We have thoroughly revised the manuscript to improve the clarity and readability of the language. We ensured that key points are communicated effectively.
We believe these revisions address your concerns and significantly improve the manuscript. Thank you once again for your constructive feedback and valuable suggestions.
Kind regards,
Marika Bielecka

Reviewer 3 Report
Comments and Suggestions for Authors
The evaluated article aimed to assess the physicochemical and sensory quality, as well as the viability of LAB, in high-protein yogurts produced using different dairy sources.
Abstract: The abstract is poorly written and needs to be reformulated. It lacks an introduction, and the results are presented in a very superficial manner. Additionally, the authors unnecessarily included information about the department where the study was conducted. I suggest restructuring the abstract to emphasize the main results more effectively.
Keywords: Use terms not already present in the title to improve indexing.
Microbiological analyses: These are only mentioned in the methodology section. There is no reference to them in the abstract, objectives, or introduction. The authors should clarify the reasons for conducting these analyses and include this information in the manuscript.
Results and Discussion Section: The current structure is somewhat confusing. I recommend renaming this section to "Results and Discussion" and separating it into two distinct parts to improve clarity.
Topics 3.1 and 3.2: These need to be better discussed and appropriately referenced.
Formatting and Terminology:
- The scientific names of microorganisms should be written in full at the first mention, and subsequently, only the genus abbreviation should be used.
- The abbreviation "CFU" should be written in uppercase letters.
Conclusions: The conclusions must be reformulated as they currently repeat the results. They should be concise, precise, and focused on addressing the proposed objectives.
Minor Remarks:
- L33-44; 55-60; 61-67: These sections require proper referencing.
- L55: Review and adjust all references to comply with the journal's formatting guidelines.
- L70-74: Remove the information about the study location, as it does not add value.
- L200-202: Remove the information about subheadings that originates from the template.
- L221-227: This content belongs to the discussion, not the results section. Move it to the appropriate section.
- L274; 304; 307: The results section should not include discussions or references.
- L324: Review and revise this line for clarity.
Author Response
Dear Reviewer,
Thank you for your detailed and thoughtful review. We greatly appreciate the time and effort you invested in providing valuable feedback, which has helped us improve the quality and clarity of our manuscript. Below, we provide a point-by-point response to each of your comments, along with details of the revisions made to the manuscript.
- Abstract:
Comment: The abstract is poorly written and needs to be reformulated. It lacks an introduction, and the results are presented in a very superficial manner. Additionally, the authors unnecessarily included information about the department where the study was conducted. I suggest restructuring the abstract to emphasize the main results more effectively.
Response: Thank you for your valuable feedback. The abstract has been revised according to your suggestions. We have restructured it to improve clarity and coherence, added a brief introduction, and emphasized the main results more effectively. The unnecessary information about the department has been removed to ensure a concise and focused abstract.
- Keywords:
Comment: Use terms not already present in the title to improve indexing.
Response: Thank you for your suggestion. We have revised the keywords to include terms not already present in the title, enhancing indexing and ensuring broader discoverability.
- Microbiological Analyses:
Comment: These are only mentioned in the methodology section. There is no reference to them in the abstract, objectives, or introduction. The authors should clarify the reasons for conducting these analyses and include this information in the manuscript.
Response: Thank you for your insightful feedback. We have clarified the rationale for the microbiological analyses by adding an explanation in the introduction and referencing these analyses in the abstract. These additions provide a clearer understanding of the purpose and significance of these analyses within the study.
- Results and Discussion Section:
Comment: The current structure is somewhat confusing. I recommend renaming this section to "Results and Discussion" and separating it into two distinct parts to improve clarity.
Response: Thank you for your suggestion. We have renamed the section to "Results and Discussion." However, after careful consideration, we believe that combining the results and discussion maintains the coherence and conciseness of the manuscript. Separating the two may unnecessarily increase the length and disrupt the flow. We hope this approach meets your expectations.
- Topics 3.1 and 3.2:
Comment: These need to be better discussed and appropriately referenced.
Response: We appreciate your feedback. The discussion in Section 3.1 has been expanded to provide a more thorough analysis of the chemical composition of the yogurts, with additional references included to support our findings. In Section 3.2, we have elaborated on the protein profile, specifically focusing on the casein-to-whey protein ratio, and included relevant citations to strengthen the discussion.
- Formatting and Terminology:
Comment:
- The scientific names of microorganisms should be written in full at the first mention, and subsequently, only the genus abbreviation should be used.
- The abbreviation "CFU" should be written in uppercase letters.
Response: Thank you for highlighting these issues. We have ensured that the scientific names of microorganisms are fully written out at their first mention and abbreviated thereafter. Additionally, "CFU" is consistently formatted in uppercase throughout the manuscript.
- Conclusions:
Comment: The conclusions must be reformulated as they currently repeat the results. They should be concise, precise, and focused on addressing the proposed objectives.
Response: Thank you for your valuable suggestion. We have revised the Conclusions section to be concise and focused on the study’s objectives, avoiding repetition of the results. The revised section emphasizes the key findings and their implications.
- Minor Remarks:
- L33-44; 55-60; 61-67: These sections require proper referencing.
Response: References have been added to these sections to ensure proper citation and compliance with the journal's guidelines. - L55: Review and adjust all references to comply with the journal's formatting guidelines.
Response: All references have been reviewed and adjusted to align with the journal's formatting requirements. - L70-74: Remove the information about the study location, as it does not add value.
Response: The information about the study location has been removed. - L200-202: Remove the information about subheadings that originates from the template.
Response: The placeholder text has been removed. - L221-227: This content belongs to the discussion, not the results section. Move it to the appropriate section.
Response: The content has been relocated to the discussion section. - L274; 304; 307: The results section should not include discussions or references.
Response: Discussions and references in these lines have been moved to the appropriate section. - L324: Review and revise this line for clarity.
Response: This line has been revised for improved clarity. - General Comment: Wiersz został usunięty.
Response: The specified line has been removed.
We believe these revisions comprehensively address your comments and significantly enhance the quality and clarity of the manuscript. Thank you once again for your constructive feedback and support.
Sincerely,
Marika Bielecka

Round 2
Reviewer 2 Report
Comments and Suggestions for Authors
This paper has reached publication level.
Comments on the Quality of English LanguageThe English could be improved to more clearly express the research.
Author Response
Dear Reviewer,
Thank you for your constructive feedback on our manuscript entitled "Effect of Fortification with High Milk Protein Preparations on Yogurt Quality." We have carefully considered your comments and made the appropriate revisions to improve the quality and clarity of the manuscript.
Comment:
The English could be improved to more clearly express the research.
Response:
Thank you for your feedback regarding the language quality of the manuscript. In response to your observation, the text has undergone a comprehensive revision to enhance clarity and readability. All language improvements have been implemented and marked in the revised manuscript for your convenience. If there are specific areas that you believe require further attention, we kindly ask for your detailed input so we can address them more effectively.
We hope these changes meet your expectations and contribute to a clearer presentation of the research findings.
Sincerely,
Marika Bielecka

Reviewer 3 Report
Comments and Suggestions for Authors
Manuscript was improved.
Author Response
Dear Reviewer,
Thank you for your thoughtful evaluation of our revised manuscript entitled "Effect of Fortification with High Milk Protein Preparations on Yogurt Quality."
We sincerely appreciate your positive feedback and acknowledgment of the improvements made in response to the previous reviews. Your comments affirm our efforts to enhance the manuscript's quality and clarity.
If there are any additional suggestions or aspects you feel require further attention, we would be happy to address them.
Thank you once again for your valuable input and for contributing to the development of our work.
Sincerely,
Marika Bielecka
